# The Effect of In-Office Bleaching with Different Concentrations of Hydrogen Peroxide on Enamel Color, Roughness, and Color Stability

**DOI:** 10.3390/ma16041389

**Published:** 2023-02-07

**Authors:** Hanife Altınışık, Sinem Akgül, Merve Nezir, Suat Özcan, Esra Özyurt

**Affiliations:** 1Department of Restorative Dentistry, Faculty of Dentistry, Gazi University, Ankara 06510, Turkey; 2İzmir Training Dental Hospital, İzmir 35160, Turkey

**Keywords:** hydrogen peroxide, office bleaching, color stability, esthetic dentistry, surface roughness, AFM

## Abstract

The aim of this study is to evaluate the effectiveness of in-office bleaching in esthetic dentistry on the roughness and color stability of the enamel surface, using different concentrations of hydrogen peroxide (HP). Fifty human incisors were randomly divided into 5 groups (*n* = 10). No bleaching was performed in the control group. For these groups, concentrations of 40% HP with fluoride (F), 35% HP with calcium (Ca), 25% HP with nano-hydroxyapatite (nHA) and 18% HP with nHA were used for bleaching in the test groups. Surface roughness was assessed at baseline after bleaching occurred. Color measurements were first obtained at baseline, then after the first and second sessions of bleaching, and, finally, after the staining protocol. Scanning electron microscopy and atomic force microscopy were performed. Statistical analysis was conducted with a one-way ANOVA, followed by a post hoc Tukey’s test and a paired-samples *t*-test (*p* < 0.05). All the bleaching gels used exhibited a similar color change (*p* > 0.05). Bleaching gels containing 18% HP with nHA and that containing 35% HP with Ca caused less surface roughness of the enamel. Of these concentrations, 25% HP with nHA caused the most surface roughness and no significant difference was observed, compared with 40% HP with F. The highest coloration after bleaching was observed in 40% HP with F and 25% HP with nHA. The lowest coloration was obtained in 35% HP with Ca and 18% HP with nHA but no significant difference was observed between them and the control group. A concentration above 18% HP does not increase the bleaching effectiveness. The results show that 18% HP with nHA and 35% HP with Ca resulted in the least increase in enamel surface roughness when compared to high-concentrate HP; however, it also prevented recoloration after bleaching.

## 1. Introduction

In recent years, the demand for esthetic dentistry applications has increased with the increasing concerns experienced by patients about their appearance. An important part of aesthetic dentistry constitutes tooth bleaching. Tooth bleaching is considered to be the most effective, safe, and conservative method used to treat tooth discoloration [1]. Hydrogen peroxide (HP) and carbamide peroxide (CP) are commonly used chemicals for bleaching. During in-office bleaching, high-concentrate HP (20–45%), low-concentrate HP (<20%), or high-concentrate CP (≥37%) are used. When these oxidizing agents are applied to the tooth surface, they rapidly diffuse into the enamel and dentin and then break down to produce unstable free radicals. Bleaching of the tooth is achieved when these free radicals attack organic substances and the pigmented molecules become less reflective of light [2].

While HP whitens teeth, it also diffuses rapidly into the tooth structure due to its low molecular weight, reaches the pulp chamber, and then causes tooth sensitivity due to the release of inflammatory mediators [3,4]. In addition, HP causes microscopic changes such as increased porosity, depression, and surface irregularities [5], surface roughness increase [6], and surface hardness decrease [7]. These adverse effects occur, depending on the composition of the bleaching gels, the concentration of the peroxide, pH, bleaching technique protocols, and application time [8]. The addition of remineralizing compounds [7,9] and the reduction of peroxide concentrations in bleaching gels [9,10] avoid these adverse effects caused by high-concentrate HP. Studies have reported that the addition of remineralizing compounds to office bleaching gels containing highly concentrated HP causes an increase in enamel microhardness [7] and a decrease in enamel erosion [11]. A recent study demonstrated effective bleaching performance with low-concentrate HP bleaching gels [12]. However, in other in vitro studies, less cell damage [13] and fewer surface changes at these low peroxide concentrations were reported [10].

Manufacturers have introduced a wide range of bleaching agent concentrations and application techniques to improve the bleaching effect. As a result, there has been a significant increase in commercially available in-office tooth-bleaching products that commonly use high levels of hydrogen (15% to 45%) or carbamide (≥37%) peroxides. Considering that peroxide concentration and the application technique are the main factors that can affect the effectiveness of bleaching agents, the aim of this study was to evaluate the effects of four hydrogen peroxide concentrations commonly used in bleaching treatments on color change, surface roughness, and color stability. Three null hypotheses were tested: (I) the effectiveness of bleaching does not increase with the concentration of HP; (II) HP concentration does not affect the enamel surface roughness; (III) color stability does not change with HP concentration.

## 2. Materials and Methods

The study was approved by the Ethics Committee of the Trakya University Faculty of Dentistry, under protocol no: 2021/458. G*Power 3.1 software was used to determine the minimum sample size, according to these parameters: 95% statistical significance, 0.80 test power, 0.40 effect size, and 5 experimental groups. This resulted in a minimum sample size of 50 specimens (*n* = 10/group). Fifty recently extracted human incisor teeth without cracks or enamel defects (at 40× magnification) were used in this study. The crown was separated from the root using a water-cooled low-speed diamond bur, the pulp tissue was cleaned, and each tooth was embedded in self-curing acrylic resin inside a ring model. Then, the enamel surfaces were sanded with 600–800–1000–1200 grit SIC paper under water cooling (Mecapol P230, Press, Grenoble, France) to standardize the enamel surface and create parallel planar surfaces [14]. All specimens were placed in an ultrasonic bath for 30 min after polishing to remove residues from the SIC abrasive papers.

### 2.1. Treatment Protocols

The specimens were randomly divided into 5 groups, according to the appropriate bleaching protocols. All protocols were performed according to the manufacturer’s instructions. The products evaluated in this study are shown in Table 1.

The bleaching gel application groups are as follows:Control: No bleaching was performed;40% HP with F: An in-office bleaching gel containing 40% hydrogen peroxide and fluoride (F) (Opesence Boost, Ultradent, South Jordan, UT, USA) was applied twice and each application time was 20 min, at each session;35% HP with Ca: An in-office bleaching gel containing 35% hydrogen peroxide and calcium (Ca) (Whiteness HP Blue, FGM, Santa Catarina, Brasil) was applied once for 40 min;25% HP with nHA: An in-office bleaching gel containing 25% hydrogen peroxide and nanohydroxyapatite (nHA) (Biowhiten, Biodent Ltd., İstanbul, Turkey) was applied three times and each application time was 15 min;18% HP with nHA: In-office bleaching containing 18% hydrogen peroxide and nano-hydroxyapatite (Bİowhiten, Biodent Ltd., İstanbul, Turkey) was applied five times and each application time was 10 min.

All bleaching protocols were completed by the same operator in two sessions that were 7 days apart. After each treatment session, all specimens were washed under water, then stored individually in distilled water at 37 °C. In the control group, specimens were kept in distilled water at 37 °C for two weeks and the solution was changed daily.

### 2.2. Staining Protocol

After the bleaching protocols were completed, the specimens were immersed in coffee prepared with 2 g of instant coffee powder (Nescafé Classic, Nestlé, Vevey, Switzerland) and 200 mL of boiling distilled water for 30 min per day for two weeks [15]. The solution was prepared freshly every day. Each specimen was immersed individually in 20 mL of coffee for 30 min and subsequently washed with distilled water, then immersed in distilled water during the entire day. This cycle was repeated for 2 weeks.

### 2.3. Color Measurement

Color measurements were performed using the Vita Easyshade 5 spectrophotometer (Vita Zahnfabrik, Bad Sackingen, Germany), in accordance with the CIELAB system. The spectrophotometer was calibrated in compliance with the manufacturer’s instructions before each measurement. The measurements were performed after baseline (T_0_), after the first session of bleaching (T_1_), the second session of bleaching (T_2_), and the staining protocol (T_3_). The baseline is defined as the time point before bleaching protocols were applied. The color differences between T_1_ − T_0_, T_2_ − T_0_, T_3_ − T_2_ were represented by ΔE_1_* ΔE_2_*, and ΔE_3_*, respectively, and were calculated as follows:


ΔE* = [(ΔL*)^2^ + (Δa*)^2^ + (Δb*)^2^]^1/2.^


### 2.4. Surface Roughness Measurements

The surface roughness measurements were performed in three different directions on the surface of each specimen’s enamel surface, using a profilometer device (Surface SJ-301, Mitutoyo, Kawasaki, Japan). The profilometer was calibrated with a cut-off of 0.25 mm, a reading length of 1.25 mm, and a velocity of 0.5 mm/s. The measurements were averaged and recorded as the surface roughness (R_a_) values. The surface roughness measurements were calculated after the baseline and the second session of bleaching.

### 2.5. Atomic Force Microscopy (AFM)

One specimen for each group was prepared for surface topography analysis, using a high-performance atomic force microscope (NanoMagnetics Instruments Ltd., Oxford, UK). In the bleaching treatment groups, AFM examinations were performed after a second session of the treatments. The topography of the specimens was obtained from a 40 × 40 μm^2^ area, using a tip scanning in contact mode with an average scan speed of 0.5 μm/s.

### 2.6. Scanning Electron Microscopy (SEM)

One specimen for each group was prepared for qualitative analysis of the surface morphology via scanning electron microscope (SEM) (FE-SEM, Hitachi SU5000, Hi-Tech Ltd., Tokyo, Japan). The specimen surfaces were subjected to gold sputtering. Images were acquired at X2500 magnification, at an electrical voltage of 10 kV, from the most representative areas of the specimens. In the bleaching treatment groups, SEM examinations were performed after the second session of treatments.

### 2.7. Statistical Analysis

The color measurements and roughness values were analyzed with a one-way analysis of variance (ANOVA) and post hoc Tukey test. For the surface roughness evaluation, while comparing samples before and after the application of bleaching gel in each group, a paired-sample *t*-test was used. All tests were carried out using the IBM SPSS Statistics 22 software (IBM Corporation, Armonk, NY, USA) with a level of significance of 5%.

## 3. Results

The color-change results of all groups are presented in Table 2. A graphical comparison is also shown in Figure 1. After the first session and the second session of bleaching, the control groups were evaluated among themselves; in the control group, statistically significantly lower values were obtained compared with those of the bleaching groups (*p* < 0.05). However, no significant differences in color change were observed between the treatment groups. After the staining protocol, a post hoc Tukey HSD test indicated that the 40% group with F showed significantly different results from the control, while the 35% group with Ca and the 25% group with nHA exhibited significant differences from the control, as did the groups of 35% with Ca and 18% with nHA.

The surface roughness values of the groups after evaluation are summarized in Table 3. Accordingly, no significant difference was obtained in any group between the samples before and after the bleaching gel applications, except in the control group (*p* = 0.067 in the control group). In the groups before the application of bleaching gel, there were no significant differences among the tested groups (*p* > 0.05). In the groups after the application of bleaching gel, according to the post hoc Tukey test results, both the samples of 40% HP with F and 25% HP with nHA showed statistically significant high values compared with the control samples, as with the samples of 35% HP with Ca and 18% HP with nHA (*p* = 0.000). Between 40% HP with F and 25% HP with nHA, and between control, 35% HP with Ca and 18% HP with nHA there was no significant difference was observed (*p* > 0.05).

Figure 2A–E shows the enamel morphology before and after the application of bleaching gels with different HP concentrations. In Figure 2A, the micrographs of unbleached enamel (control) are reported; smooth surface morphology, the presence of aprismatic enamel, and a small incidence of porosities can be observed. Conversely, in all bleaching-applied enamel specimens (Figure 2B–E), we observed different types of defects, and a distinct severity of such events was found. In enamel bleached with 40% HP (Figure 1B) and 25% HP (Figure 2D), we observed serious surface changes, such as the complete removal of the aprismatic layer and an increased depth of enamel irregularities. More dissolution was observed in the prism cores in Figure 1D than in Figure 1B. In enamel bleached with 35% HP (Figure 2C) and 18% HP (Figure 2E), we observed the partial removal of the aprismatic layer and fewer enamel irregularities.

Figure 3A–E presents representative AFM images of the enamel surfaces for the control and for samples following the application of bleaching gel. In AFM images, increased surface irregularities were observed for all bleached enamel specimens (Figure 3B–E) compared with the control group (Figure 3A). After bleaching with 40% HP (Figure 3B) and 25% HP (Figure 3D), more irregularities and deeper enamel grooves were observed on the enamel surface, while less irregularity was observed after bleaching with 35% HP (Figure 3C) and 18% HP (Figure 3E).

## 4. Discussion

The first null hypothesis investigated in this study was accepted, as all office bleaching gels produced a similar color change after both the first session and the second session, regardless of the HP concentration. Table 2 shows the color change results in the tested groups.

To assess changes in tooth shade after bleaching, subjective methods are used, such as a comparative visual analysis with a standard tooth-color guide, as well as objective methods, such as the use of spectrophotometers, colorimeters, and digital cameras [16]. Compared to other devices, spectrophotometers have a few advantages, such as being more practical than digital cameras and more accurate than colorimeters [17]. Regarding the use of spectrophotometers for in vitro studies, there are conflicting opinions. According to one study, the Vita Easyshade is a clinical device that is typically not advised for use in in vitro testing because it only operates in “tooth mode” [18]. However, other studies have reported that it provides reliable color measurements under both laboratory and clinical conditions [18,19]. Therefore, the Vita Easyshade was used to determine the effectiveness of the bleaching agents used in this study.

Studies have reported that the use of high-concentrate HP does not show any additional benefit in terms of color change when compared with low-concentrate HP [20,21,22]; this finding is consistent with the results of our study. Kawamoto et al. [23] reported that the number of free radicals in peroxide solutions was related to concentration, so that the bleaching efficiency increased as the HP concentration in the bleaching gels increased. However, some clinical studies evaluating the efficacy of office bleaching gels at different concentrations have also reported that the whitening effect is not only concentration-dependent [24,25,26]. Application time and period are more efficient than the concentration of HP in the final shade changes [27]. Thus, alternative approaches are used to increase the effectiveness of in-office bleaching gels containing low concentrations of HP, such as increasing the number of changes or the application time in one session of bleaching gels [25], using them when combined with the home bleaching technique [26], and the activation of bleaching gels with light [24].

Generally, in-office bleaching gels containing HP are applied to the tooth surfaces for from 5 to 20 min and are left undisturbed during this time. Depending on the rapid degradation of the HP, it could not oxidize the organic content of dentin. Therefore, the repeated application of bleaching gel in two or five iterations during each appointment is recommended, according to the manufacturer. Studies have reported that the concentrations of the active ingredient in bleaching gels decreased linearly with time [28,29]. Kose et al. [25] reported higher bleaching efficiency when the in-office bleaching gel was renewed several times. In a clinical study, Reis et al. [30] demonstrated that three separate 15-minute applications of 35% HP allowed more bleaching than a single 45-minute application. Ito et al. [31] reported that pH conditioners influenced whitening efficacy, and the degree of this effect was monitored in sodium bicarbonate (NaHCO_3_), potassium hydroxide (KOH), and sodium hydroxide (NaOH). In another study, tooth-whitening systems containing sodium bicarbonate showed more efficacy than systems without sodium bicarbonate in removing internal stains [32]. The number of changes and the application times for 18% HP with nHA bleaching gels are greater than with highly concentrated HP gels. Sodium bicarbonate and sodium hydroxide, used as pH conditioners, exist in 25% HP with nHA and 18% HP with nHA, and potassium hydroxide is used with 40% HP with F. The pH conditioner used with 35% HP with Ca was not disclosed. In addition, according to the manufacturers, all the bleaching gels used in this study had an initial pH > 7.0. However, the pH values claimed by the manufacturer were not confirmed when measuring the pH in this study. Xu et al. [33] reported that neutral or alkaline bleaching gel yielded more obvious bleaching efficacy, without any evidence of enamel erosion. For all these reasons, there is no difference between the final color changes with the low-concentrate HP bleaching gel and the highly concentrated HP bleaching gels.

Another purpose of this study was to evaluate the effects of different concentrations of HP in the in-office bleaching gels on enamel surface roughness, using profilometry, scanning electron microscopy (SEM), and atomic force microscopy (AFM). Profilometry is generally preferred when evaluating the effects of bleaching gels on enamel surface roughness. However, imaging technologies such as SEM and AFM can also be used to conduct more detailed investigations. In Table 3, it can be seen that HP increases the surface roughness of the enamel, regardless of the concentration. Although the group with the greatest roughness change on the enamel surface was 25% HP with nHA, there was no significant difference between the surface roughness of 40% HP with F and 25% HP with nHA. Additionally, no significant difference between the roughness changes of 35% HP with Ca and 18% HP with nHA was recorded. The SEM and AFM images also supported these surface roughness values. According to these results, the second hypothesis of the study was rejected. In a study where specimens were stored in artificial saliva during the experiment, the authors reported that the calcium and phosphate ions lost as a result of bleaching can be recovered through ion exchange from saliva [34]. Studies using distilled water as a storage solution have reported that no ion exchange was exhibited in dental hard tissues [35]. For this reason, in our study, the specimens were kept in distilled water instead of artificial saliva to detect the side effects of different concentrations of HP in the in-office bleaching gels more clearly. The peroxide content of bleaching gels reduces the mineral content of the enamel. As the minerals are separated from the surface, this causes deformations in the nuclei of the enamel prism and the interprism areas, the surface hardness decreases, and the roughness increases [36]. The effectiveness of bleaching gels and the changes to the enamel surface are mainly affected by concentration, composition, exposure time, and the diffusion capacity and pH of the bleaching gels [37,38,39]. Reapplication and longer exposure times also increased HP penetration [40]. Kwon et al. [41] reported that the viscosity of the bleaching gel did not influence bleaching efficacy, but lower-viscosity gels increased hydrogen peroxide penetration. Although no significant increase in surface roughness values after bleaching with 40% HP was reported in the study by Kolsuz et al. [42], in another study, a significant difference was observed in the surface roughness values, which is in line with the findings of the current study [43]. Here, the gels with 25% HP with nHA and 18% HP with nHA had lower viscosity than the other bleaching gels evaluated in the study. A concentration of 25% HP with nHA may have penetrated more deeply into the tooth due to its low viscosity, as well as due to the application time and the number of changes. For this reason, it caused dissolution in both the nuclei of the enamel prism and the interprism areas. The results of treatment with 25% HP with nHA could not be compared with the results from other studies since there is no published study on changes in the tooth surface. The addition of calcium compounds to bleaching gels helps to reduce the damage to the microhardness of the enamel caused by the bleaching and prevent mineral loss, thus reducing tooth sensitivity. In addition, calcium compounds maintain a stable pH throughout the bleaching process [44]. For these reasons, although 35% HP with Ca contains a high concentration of HP, the gel caused less surface roughness than other high-concentrate HP gels, due to the single application number and the protective effect of Ca. Although the 18% HP gel contained nHA and a low concentration of HP, it caused a little surface roughness on the tooth enamel, due to its high application time, the number of changes, and its low viscosity. Grazioli et al. [10] stated that the bleaching gel containing 15% HP caused less decrease in enamel surface hardness and did not show a surface alteration to the enamel, compared to the 25–35% HP gels. They also found that 15% HP was the maximum effective concentration to prevent morphological changes in the enamel; higher concentrations did not improve the whitening effect but did increase the possibility of enamel damage.

As the pores on the enamel surface increase after bleaching, the precipitation of various color pigments into these pores also increases. This will accelerate the color-change process of the tooth surface after bleaching [45]. Previous studies reported that the addition of Ca, F, and nHA in gels reduced enamel demineralization during bleaching treatments [46,47]. Vieira et al. [48] reported that 40% HP with F did not reduce the enamel mineral loss since high-concentrate HP reduced the effects of F. In the same study, it was reported that 35% HP with Ca may prevent enamel mineral loss, since Ca can move through the tooth structure during demineralization events by diffusion. In this study, it was observed that the most marked color change was exhibited in those specimens that were kept in coffee after bleaching with 40% HP and 25% HP. This may be a result of the increasing effect of these gels on enamel surface roughness, compared to the others. Overall, 18% HP with nHA and 35% HP with Ca exhibited a slight increase in surface roughness compared to the control group, although there was no significant difference in color stability. Therefore, the third null hypothesis of this study was partially rejected.

Finally, it would be appropriate to list the limitations of this in vitro study. In vitro studies cannot fully imitate clinical conditions, due to their nature. In addition, because the teeth included in the study were randomly selected, the effect of previous conditions, such as dental age and possible interventions, is unknown. A further limitation is that saliva is not used as a remineralizing agent following in-office bleaching, because saliva plays an important role in protecting the enamel from mineral loss and allows enamel remineralization [10,34]. Vargas et al. [49] report that saliva acts as an alkalizing agent and helps to reverse the effects of bleaching. However, in this study, the specimens were stored in distilled water instead of in artificial saliva during the study period, to analyze the effect of bleaching agents on the surface morphology of the enamel. With the increasing demand for dental bleaching and the development of new formulations, the introduction of bleaching gels containing different concentrations of HP and different remineralizing agents, such as Ca, F, and nHA, has allowed bleaching treatments that can provide excellent esthetic results without tissue damage. However, further in vitro, in situ, and in vivo studies are required to evaluate the bleaching efficacy and enamel surface effects of these products.

## 5. Conclusions

Within the limitations of the present study, bleaching gels containing 18% HP with nHA and 35% HP with Ca showed similar color changes and lower enamel surface roughness values compared to the other bleaching gels used in this study. The results of this study show that combined products with low HP concentrations, pH gels (neutral to alkaline), and short exposure time can be suggested for the purposes of bleaching. Although high concentrations of HP will be used, those with Ca-added gels may be preferred, to reduce the effect of treatment on the tooth surface. The optimal interaction of peroxide concentration, content, exposure time, and the mode of application is essential for an optimal bleaching treatment with no or few side effects.

## Figures and Tables

**Figure 1 materials-16-01389-f001:**
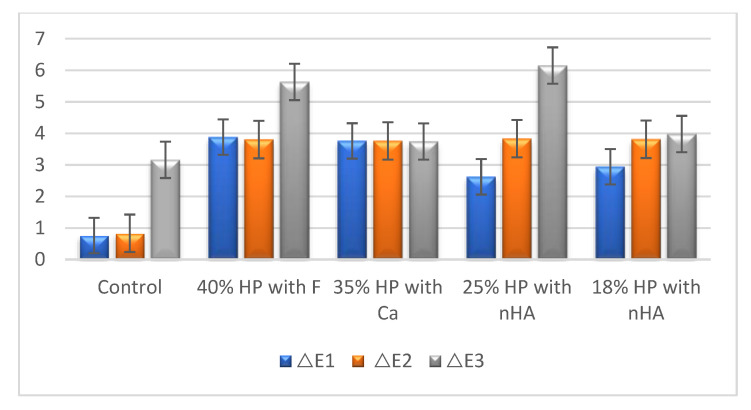
Comparison of the ΔE values of the groups after the bleaching and staining protocols: ΔE1 is after the first session of bleaching; ΔE2 is after the second session of bleaching; ΔE3 is after the staining protocol. HP, hydrogen Peroxide; F, fluoride; Ca, calcium; nHA, nano-hydroxyapatite.

**Figure 2 materials-16-01389-f002:**
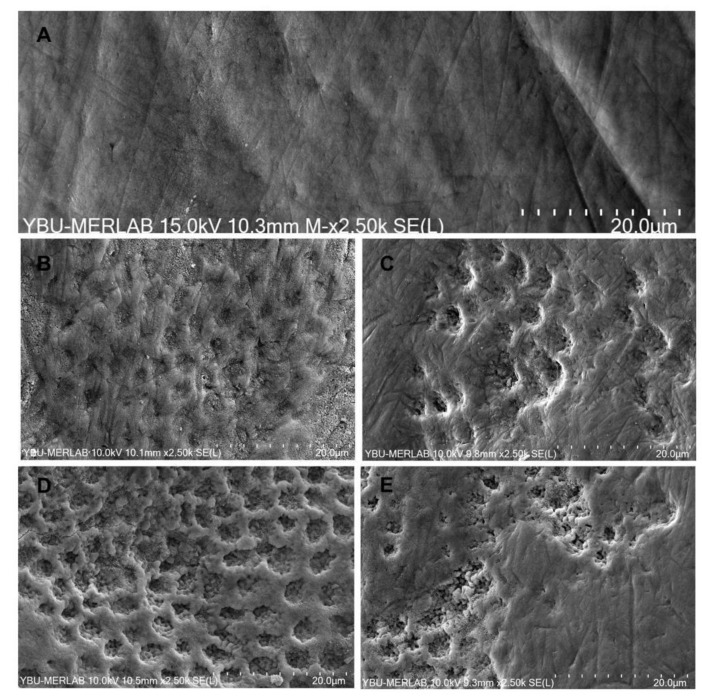
Scanning electron microscopy photographs of enamel surfaces before and after the application of bleaching gels with different HP concentrations: (**A**) no bleaching (control); (**B**) bleaching with 40% HP + F; (**C**) bleaching with 35% HP + Ca; (**D**) bleaching with 25% HP + nHA; (**E**) bleaching with 18% HP + nHA.

**Figure 3 materials-16-01389-f003:**
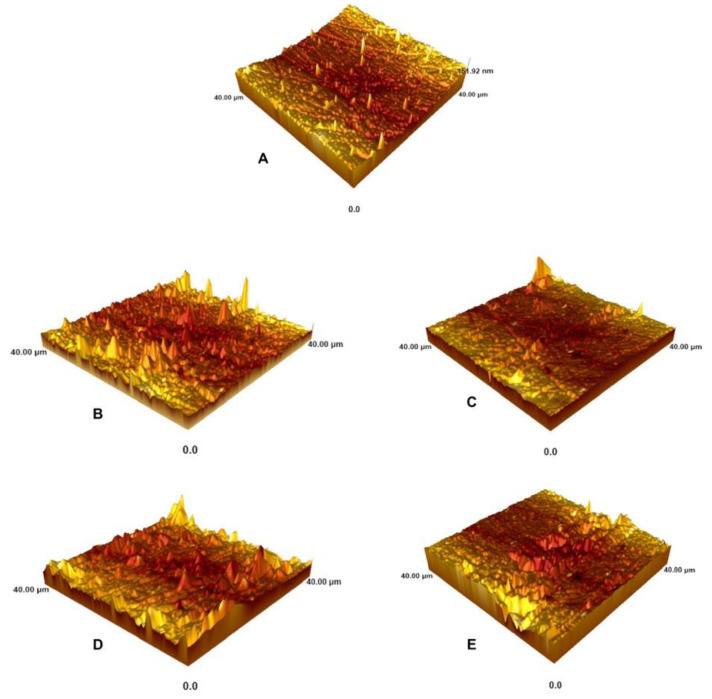
Three-dimensional atomic force microscopy images before and after the application of bleaching gels with different HP concentrations: (**A**) no bleaching (control); (**B**) bleaching with 40% HP + F; (**C**) bleaching with 35% HP + Ca; (**D**) bleaching with 25% HP + nHA; (**E**) bleaching with 18% HP + nHA.

**Table 1 materials-16-01389-t001:** Bleaching gels used in this study.

Product, Manufacturer, Batch Number	Abbreviation	Application Protocol	Total Time	pH Indicated by the Manufacturer	Active Principle (Commercial Presentation)	Ingredients (Technical Profile)
Opalescence Boost, Ultradent, South Jordan, UT, USA	40% HP with F	2 applications of 20 min per session	40 min	Neutral (pH = 7)	Hydrogen Peroxide 40% and fluoride	Hydrogen Peroxide 40%, Potassium Nitrate, Potassium Hydroxide, Sodium Fluoride, Dimethicone and Glycerin.
Whiteness HP Blue, FGM, Santa Catarina, Brazil	35% HP with Ca	1 application of 40 min per session	40 min	Alkaline and stable (pH = 8–9)	Hydrogen Peroxide 35% and calcium	Hydrogen Peroxide 35%, thickeners, violet pigment, neutralizing agents, calcium gluconate, glycol, and deionized water.
Biowhiten, Biodent Ltd., İstanbul, Turkey	25% HP with nHA	3 applications of 15 min per session	45 min	Alkaline (pH ≥ 7.5)	Hydrogen Peroxide 25% and nano-hydroxyapatite	Water, Glycerin, Alcohol, Sodium bicarbonate, Sodium hydroxide, 25% HP and nHA
Biowhiten, Biodent Ltd., İstanbul, Turkey	18% with nHA	5 applications of 10 min per session	50 min	Alkaline (pH ≥ 7.5)	Hydrogen Peroxide 18% and nano-hydroxyapatite	Water, Glycerin, Alcohol, Sodium bicarbonate, Sodium hydroxide, 18% HP and nHA

Abbreviations: HP, hydrogen Peroxide; F, fluoride; Ca, calcium; nHA, nano-hydroxyapatite.

**Table 2 materials-16-01389-t002:** Mean and standard deviations of ΔE values for the treatment groups after the bleaching and staining protocols.

Treatment Groups	After the First Session of Bleaching	After the Second Session of Bleaching	After the Staining Protocol
	ΔE1	ΔE2	ΔE3
Control	0.76 ± 0.16 ^a^	0.83 ± 0.19 ^a^	3.16 ± 1.24 ^a^
40%HP with F	3.88 ± 1.63 ^b^	3.80 ± 1.64 ^b^	5.63 ± 1.2 ^b^
35%HP with Ca	3.76 ± 1.76 ^b^	3.76 ± 1.55 ^b^	3.74 ± 1.35 ^a^
25%HP with nHA	2.62 ± 1.24 ^b^	3.83 ± 1.51 ^b^	6.15 ± 1.5 ^b^
18%HP with nHA	2.94 ± 1.50 ^b^	3.81 ± 1.52 ^b^	3.98 ± 1.20 ^a^
*p*	0.000 *	0.000 *	0.000 *

One-way ANOVA test * *p* < 0.05. Different letters in the columns show the statistical difference (*p* < 0.05).

**Table 3 materials-16-01389-t003:** Comparison of the mean and standard deviation values of surface roughness for the treatment groups before and after bleaching.

Treatment Groups	Before Bleaching	After Bleaching	*p* ^2^
Control	0.23 ± 0.04 ^a^	0.24 ± 0.04 ^a^	0.067
40%HP with F	0.24 ± 0.07 ^a^	0.48 ± 0.11 ^b^	0.000 *
35%HP with Ca	0.23 ± 0.05 ^a^	0.31 ± 0.06 ^c^	0.000 *
25%HP with nHA	0.21 ± 0.06 ^a^	0.55 ± 0.06 ^b^	0.000 *
18%HP with nHA	0.22 ± 0.06 ^a^	0.33 ± 0.07 ^c^	0.000 *
*p* ^1^	0.716	0.000 *	

^1^  *p* represents the one-way ANOVA test results; ^2^  *p* represents the paired-sample *t*-test results. Different letters in the same column and row show statistical significance (* *p* < 0.05).

## Data Availability

Not applicable.

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
