# Peer review of "The Effect of In-Office Bleaching with Different Concentrations of Hydrogen Peroxide on Enamel Color, Roughness, and Color Stability"

_materials, 2023, doi:10.3390/ma16041389_

Round 1

Reviewer 1 Report

This paper is about the effect of in-office bleaching with different concentrated hydrogen peroxide on enamel color, roughness, and color stability.

The paper structure and the overall content are good, but I notice a big bias (see below). 

Nevertheless, I suggest some improvements be performed before this manuscript can be considered suitable for publication.

Line 64: The authors wrote:

“Then, the enamel surfaces were sanded with 600-800-1200 grit SIC paper under water cooling (Mecapol P230, Press) to standardize the enamel surface. ”

The reviewer thinks that this could be a bias rather than a standardization. Bleaching is performed on not polished surfaces. Simple cleaning should have been performed. Do the authors have a reference to cite that used this approach? The removal of the aprismatic enamel raises some concerns.

The authors should have evaluated the color change induced by bleaching agents on sound teeth.

Other groups with the finishing and polishing procedures should have been used only for roughness evaluation after bleaching treatments, not for color change analysis.

Line 89: Please specify Name and brand of coffee powder

Please cite a reference for standard staining procedures: doi: 10.1111/jerd.12912.  PMID: 35396818.

  1. Please add a limitation of the Spectrophotometer used:

Easyshade (Vita Zahnfabrik, Bad Säckingen, Germany)  is a clinical device (only working in "tooth mode") and it is generally not recommended for in-vitro testing. (you could cite: https://doi.org/10.1016/j.jdent.2022.104223)

Table 2: the readers could also benefit from a bar graph (with â–³E and standard deviation) to get visual quick knowledge about the color change.

Please add other limitations of this study at the end of the discussion.

Author Response

Point 1: Line 64: The authors wrote:

“Then, the enamel surfaces were sanded with 600-800-1000-1200 grit SIC paper under water cooling (Mecapol P230, Press) to standardize the enamel surface. ”

The reviewer thinks that this could be a bias rather than a standardization. Bleaching is performed on not polished surfaces. Simple cleaning should have been performed. Do the authors have a reference to cite that used this approach? The removal of the aprismatic enamel raises some concerns. The authors should have evaluated the color change induced by bleaching agents on sound teeth. Other groups with the finishing and polishing procedures should have been used only for roughness evaluation after bleaching treatments, not for color change analysis.

Response 1: Dear reviewer; thank you for your valuable comments. In most in vitro studies where color change, surface roughness, and hardness were evaluated together, sanding with SIC paper was performed under water cooling to standardize the enamel surfaces and create parallel planar surfaces. Some of the studies in the literature are as follows. One of these articles was referenced and highlighted (Line 73).

*Goyal K, Saha SG, Bhardwaj A, Saha MK, Bhapkar K, Paradkar S: A comparative evaluation of the effect of three different concentrations of in-office bleaching agents on microhardness and surface roughness of enamel - An in vitro study. Dent Res J (Isfahan) 2021, 18:49.

* de Carvalho AC, de Souza TF, Liporoni PC, Pizi EC, Matuda LA, Catelan A: Effect of bleaching agents on hardness, surface roughness and color parameters of dental enamel. J Clin Exp Dent 2020, 12(7):e670-e675.

* Grazioli G, Valente LL, Isolan CP, Pinheiro HA, Duarte CG, Münchow EA: Bleaching and enamel surface interactions resulting from the use of highly-concentrated bleaching gels. Arch Oral Biol 2018, 87:157-162.

Point 2: Line 89: Please specify Name and brand of coffee powder

Please cite a reference for standard staining procedures: doi: 10.1111/jerd.12912.  PMID: 35396818.

Response 2: Dear reviewer, the name and brand of coffee powder used in the present study was specified and a reference for standard staining procedures was added. Changes made were highlighted. (Line 100-101)

Point 3: Please add a limitation of the Spectrophotometer used:

Easyshade (Vita Zahnfabrik, Bad Säckingen, Germany)  is a clinical device (only working in "tooth mode") and it is generally not recommended for in-vitro testing. (you could cite: https://doi.org/10.1016/j.jdent.2022.104223)

Response 3: Dear reviewer, a limitation was added for the Spectrophotometer used, the relevant article was referenced and highlighted (Line 203-213).

Point 4: Table 2: the readers could also benefit from a bar graph (with â–³E and standard deviation) to get visual quick knowledge about the color change.

Response 4: Dear reviewer, to compare the groups’ color changes more easily, the table was converted into a bar graph and added to the article as Figure 1 (Line 153-156).

Point 5: Please add other limitations of this study at the end of the discussion.

Response 5: Dear reviewer, the limitations of this study were added at the end of the discussion and highlighted (Line 308-317).

Reviewer 2 Report

This is a well written manuscript which is important for the readers.

There are some minor mistake in the manuscript 

for example: page 2  after line 46, three lines are from materials and methods.

and page 3 line 98 the first (T2) should be (T1)

I think it would be better to add limitations of study at the end of discussion

regards

Author Response

Point 1: There are some minor mistake in the manuscript for example: page 2  after line 46, three lines are from materials and methods and page 3 line 98 the first (T2) should be (T1)

Response 1: Dear reviewer; thank you for your valuable comments. Relevant mistakes have been corrected and highlighted in the article (Line 110), (Line 76-78).

Point 2: I think it would be better to add limitations of study at the end of discussion

Response 2: Dear reviewer, the limitations of this study were added at the end of the discussion and highlighted (Line 308-317).

Reviewer 3 Report

The author does not describe in detail what nHA is.

The author did not make comparisons with groups treated using light for tooth whitening, this would make the paper more consistent.

Author Response

Response to Reviewer 3 Comments

Point 1: The author does not describe in detail what nHA is.

Response 1: Dear reviewer, thank you for your valuable comments. It was stated that nHA stands for nanohydroxyapatite and highlighted (Line 88-89).

Point 2: The author did not make comparisons with groups treated using light for tooth whitening, this would make the paper more consistent.

Response 2: Dear reviewer, the bleaching products we used in the study were products that could be used both with and without light. However, all products in this study were applied according to the manufacturer's instructions without light activation.

Reviewer 4 Report

This article does not have the innovation required for a Q1 journal.

the authors should be considered the following comments:

Introduction: Pleases improve it and write about the novelty of your study

Materials & Methods: Pleases add a reference for the Staining protocol

Materials & Methods: Pleases provide sample calculation

Results: Pleases add tables for the two by two comparisons tests between the various types of the bleaching treatment

Discussion: Please add a paragraph for limitations 

Author Response

Response to Reviewer 4 Comments

Point 1: Introduction: Pleases improve it and write about the novelty of your study

Response 1: Dear reviewer; thank you for your valuable comments. The sentence that improves the introduction of the article and explains the purpose of the study more clearly was added and highlighted (Line 52-59).

Point 2: Materials & Methods: Pleases add a reference for the Staining protocol

Response 2: Ple DeÄŸerli reviewer, a reference for standard staining procedures was added and highlighted. (Line 101)

Point 3: Materials & Methods: Pleases provide sample calculation

Response 3: Dear reviewer, the sentence about the calculation of the number of samples was added and highlighted (Line 64-67).

Point 4: Pleases add tables for the two by two comparisons tests between the various types of the bleaching treatment

Response 4: Dear reviewer, one-way ANOVA and post hoc Tukey test were performed to compare the effects of various bleaching products (Table 2). Different letters were used to show statistical differences between groups and highlighted (Line 150-152).

Point 5: Discussion: Please add a paragraph for limitations

Response 5: Dear reviewer, the limitations of this study were added at the end of the discussion and highlighted (Line 308-317).

Round 2

Reviewer 1 Report

The authors have amended all the requests.

Reviewer 4 Report

Dear Authors

The manuscript has been improved and it can be published.